# Disease Forecasting for the Rational Management of Grapevine Mildews in the Chianti Bio-District (Tuscany)

**DOI:** 10.3390/plants12020285

**Published:** 2023-01-07

**Authors:** Giuliana Maddalena, Elena Marone Fassolo, Piero Attilio Bianco, Silvia Laura Toffolatti

**Affiliations:** Department of Agricultural and Environmental Sciences (DISAA), University of Milan, Via Celoria 2, 20133 Milano, Italy

**Keywords:** downy mildew, powdery mildew, sustainable viticulture, organic viticulture, organic farming, sustainable viticulture, plant disease management

## Abstract

Downy and powdery mildews are major grapevine diseases. In organic viticulture, a few fungicides with protectant activities (copper and sulphur in particular) can be used, and their preventative application frequently leads to unneeded spraying. The adoption of an epidemiological disease forecasting model could optimise the timing of treatments and achieve a good level of disease protection. In this study, the effectiveness of the EPI (Etat Potentiel d’Infection) model in predicting infection risk for downy and powdery mildews was evaluated in nine organic vineyards located in Panzano in Chianti (FI), over a 2-year period (2020–2021). The reliability of the EPI model was investigated by comparing the disease intensities, the number of fungicide sprayings, the quantities of the fungicides (kg/ha), and the costs of the treatment achieved, with or without the use of the model, in a vineyard. The results obtained over two seasons indicated that, in most cases, the use of the EPI model accurately signalled the infection risk and allowed for a reduction in the frequency and cost of spraying, particularly for powdery mildew control (−40% sprayings, −20% costs compared to the farmer’s schedule), without compromising crop protection. The use of the EPI model can, therefore, contribute to more-sustainable disease management in organic viticulture.

## 1. Introduction

The vineyard system is one of the most widespread agricultural sectors in Mediterranean countries, and infections caused by fungal pathogens are one of the main causes of production losses [1]. Downy and powdery mildews are major diseases of grapevines (*Vitis vinifera* L.), with impacts on the qualitative and quantitative characteristics of grapevine production [2]. If not controlled, they can cause yield losses of up to 75–100% during a high disease-pressure year [2,3,4], and reduce wine quality [5,6,7]. Downy mildew is caused by the oomycete *Plasmopara viticola* (Berk. and Curt.) Berl. and de Toni, while powdery mildew is caused by the ascomycete *Erysiphe necator* Schwein. They are both biotrophic parasites infecting all green parts of the plant, as well as polycyclic pathogens that reproduce through sexual spores, responsible for primary infections, and asexual spores, causing secondary infections [8,9]. *P. viticola* overwinters thanks to the differentiation of oospores [10,11] that germinate in spring, producing the inoculum for primary infections (zoospores produced in macrosporangia). Subsequent infection cycles are caused by zoospores produced in sporangia. The primary source of inoculum for powdery mildew comes from overwintering chasmothecia, where ascospores are formed, and secondary infections are initiated by conidia. Conidia, generated by the mycelium overwintered in buds, are alternative sources of inoculum for primary infections. In this case, at sprouting, the mycelium produces conidia that colonise the developing shoot, causing distorted and stunted growth (flag shoots) [12]. The number of flag shoots can vary markedly between years, and they can be extremely rare or even completely absent [13]. More commonly, colonisation of a flag shoot is less extensive, and infection of a single leaf, or of leaves on only one side of a shoot, is observed [14]. Since most of the grapevine cultivars are susceptible to *P. viticola* and *E. necator*, viticulture cannot be practised without the application of plant protection products (PPP) with fungicidal activity. Disease control strategies aim to prevent pathogen infections [15]. The most common practises for controlling the diseases rely on fungicide applications from early spring onwards, in relation to weather conditions and to the geographic location of the vineyard [16,17,18]. Treatments are applied according to a very strict calendar that often does not consider whether the environmental conditions are conducive to the diseases, leading to fungicide applications that, a posteriori, were unnecessary. Within the new framework of the European Green Deal, the European Commission launched the EU Farm to Fork Strategy (https://ec.europa.eu/food/horizontal-topics/farm-fork-strategy_en accessed on 4 December 2022), which aims to reduce the overall use of chemical pesticides by 50% by 2030 and to increase organic agriculture. In 2019, the certified organic vineyard surface area across the world was estimated at 454 kha, representing 6.2% of the world total area under vines [19]. The certified global organic vineyard surface area increases by an average of 13% per year. Even though organic viticulture is practised in a total of 63 countries across all continents, the highest concentration is found in Europe. In organic viticulture, even though many alternatives, such as essential oils and plant extracts with potential antifungal activity have been proposed [20,21,22,23], copper-based fungicides have been, and still are, intensively used to prevent downy mildew [24]. The lack of penetrative capability and the curative activity of copper fungicides lead to very frequent treatments, which are often applied weekly throughout the season, to provide an adequate protection from infections [25]. Copper has been used in viticulture for more than 150 years, at rates of up to 80 kg/ha per year, which has led to the accumulation of copper in the topsoil of many vineyards [26]. The high copper concentrations in soils and water [27] cause negative impacts on non-target soil and aquatic organisms [28]. For this reason, current legislation (Regulation EU 2018/1981) strictly limits the maximal dose of copper to 4 kg/ha/year over 7 years. 

Control of powdery mildew in organic viticulture mainly consists of the fixed-interval application of sulphur, with a consequently high number of spray applications in each growing season [29], leading to unjustified sprayings. Moreover, sulphur has undesirable side-effects, such as toxicity to beneficial mites and insects, and it may also impart off-flavours which reduce the quality and value of wine [30]. Another disadvantage is that sulphur can contribute to environmental pollution [31]. 

The timing of the applications of inorganic products used to manage plant diseases is critical because they possess a protective, rather than a curative, activity [32]. The European Commission enforces national action plans for pesticide reduction, encouraging the use of monitoring networks (Directive 128/2009/EC), forecasting models, and dissemination tools to share this information among growers and technicians [33]. Disease forecasting models assist farmers in decision-making for crop protection, helping them to identify the correct timing for fungicide applications and to optimise disease management, avoiding unnecessary interventions during the growing season. In the long term, this can result in a reduction of the number of treatments [16].

Many weather-driven mechanistic models [34,35,36,37] have been developed for predicting grapevine diseases. Mechanistic models split every stage of the lifecycle of the pathogen into different state variables and regulate changes from one state to the other by rate variables or switches, depending on environmental conditions [34]. These models rely on the estimation of several parameters and require good knowledge of the biological mechanisms and the impact of environmental variables on these mechanisms [16]. An awareness of the inherent complexity of plant disease processes and their strict dependence on several factors, with the consequent impossibility of understanding all the laws that regulate the environment–pathogen–plant system, led Strizyk to develop the EPI (Etat Potentiel d’Infection), a heuristic model, in 1981 [38,39]. The EPI model was developed by relating meteorological data with real epidemic trends which have been registered since 1907–1915 in the area of Bordeaux (France). Thanks to the numerous experiments carried out subsequently on grapevine downy mildew in France and Northern Italy [40], the EPI algorithm has undergone a series of modifications and has also been enriched with the results of the research carried out on the maturation and germination of oospores [10,41]. Currently, the model is included in the Epicure system adopted by the Institut Français de la Vigne et du Vin (IFV) (https://oadex-viti.vignevin-epicure.com). To date, this model has been applied in experimental plots of vineyards which are treated following an integrated pest management (IPM) strategy, located in Northern Italy, in order to evaluate the effectiveness of the model in quantifying epidemics of downy mildew [40].

The purpose of the present study is to validate the EPI model for both downy and powdery mildews in vineyards adopting organic farming. Simulations with the EPI model were carried out in nine organic vineyards located in the province of Florence at Panzano in Chianti (Tuscany), an important Italian viticultural area, over two vegetative seasons (2020 and 2021). The outputs of the models were used to identify the infection risk, and fungicide sprayings were applied when the model signalled medium- to high-risk. The validation was carried out by comparing: (i) the disease incidence and severity on plots not treated with fungicides (NT), those treated according to the model indications (EPI), or those treated according to the farmer’s strategy (FARM); and (ii) the number of fungicide sprayings applied in EPI and FARM plots, with attention to the economic impact.

## 2. Results

The results achieved in the nine vineyards (P1–P9) are listed below.

### 2.1. Season 2020

#### 2.1.1. Weather Conditions

The weather data are reported as the average values of minimum and maximum temperatures (°C) and rainfall (mm) per vineyard. Weather conditions, and temperatures in particular, greatly differed among the nine vineyards (Figure 1A,B). In general, the temperatures registered low values until the end of April, and a gradual increase was registered in May (Figure 1A,B). Starting from the middle of June, the temperatures showed a significant increase, particularly in the maximum values. In particular, P7 and P8 showed the highest values, whereas P2 and P3 did not register a significant increase in terms of temperatures (Figure 1A). In general, the 2020 season was very dry, characterised by very low rainfall, which was mainly concentrated in the first half of June, when 100 mm were recorded (Figure 1C). Rain was almost completely absent between mid-June and late July, accompanied by a greater increase in temperatures.

#### 2.1.2. Downy Mildew

The levels of infection risks were signalled in green (low risk), orange (medium risk) and red (high risk). The EPI model provided a medium infection risk from the 20th of April in three vineyards (P1–P3) and in most of the stations starting from the 18th of May (P6–P9). From the 25th of May, the model indicated a high infection risk until the middle of June. Moreover, a medium infection risk was signalled until the 7th of July, whereas in four vineyards (P4, P5, P6, and P8) a high infection risk was indicated on the 13th of July (Figure 2).

Following the model indications, fungicides (copper for downy mildew and sulphur for powdery mildew) were applied in presence of a medium-to-high risk (orange to red) in most of the vineyards, from the 25 May to the 7 July, whereas in the farmers’ strategy, the treatments were applied until the end of July (Table 1). Compared with the average number of fungicide sprayings in the grower spraying program, two fewer treatments were performed by following the model indications, and significant differences were observed in the average number of sprayings applied in the two strategies (*p* = 0.033).

The disease symptoms were first observed between the beginning and the end of June (Figure 2). The estimation of the incubation period highlighted that the infections occurred between the 20th of April and the 20th of June, when the EPI model indicated a medium-to-high infection risk (Table 1). Low values of disease incidence (I%I) and severity (I%D) were registered in the unsprayed plots, with average values equal to 19% and 12%, respectively, on bunches (Figure 3A). Statistical analysis highlighted significant differences in the I%I and I%D average values between NT and both the EPI strategy (*p* = 0.006) and the grower’s strategy (*p* < 0.001) (Figure 3). The two treated plots (EPI and FARM) did not show statistically significant differences for wither I%I (*p* = 0.454) or I%D (*p* = 0.472) (Figure 3).

#### 2.1.3. Powdery Mildew

A medium-to-high infection risk was indicated by the EPI model in all vineyards from the 11th of May until the 8th of June (Figure 4). However, in a limited number of vineyards (P1, P2 and P9), the forecasting model signalled a medium infection risk starting from the 27th of April to the 4th of May, and the infection risk was maintained until the 15th of June in four vineyards. Other alarms were provided in the middle of July in most of the vineyards (P1–P5, P7, and P8) (Figure 4).

The treatments in the EPI strategy were applied from the 8th to the 22nd of June in most of the vineyards, whereas in the growers’ strategy, the sprayings were carried out from the 14th of May to the 7th of July (Table 2). In the two treated plots, the average number of fungicide sprayings was significantly different (*p* < 0.001); only three treatments were applied in the EPI plot compared to seven in the growers’ plot (Table 2).

The first symptoms in the field were observed between the middle of May and the 22 June, in correspondence to the mid-to-high infection risk indicated by the model (Figure 4). A late appearance of the symptoms compared to the infection risk signalled by the model was observed in P2, where the disease was observed approximately one month later than the first indication of infection risk. In the untreated plots, the field observations highlighted an average of I%D equal to 17% and an average of I%I equal to 9.6% on the bunches. The statistical analysis highlighted the absence of significant differences between the NT plots and the EPI plots for both disease severity (*p* = 0.305) and incidence (*p* = 0.380) (Figure 5); a sporadic contamination of bunches was observed in both plots. In the EPI plots, I%I and I%D averaged 6.6% and 12.6% on bunches, respectively. Compared to the NT plots, the growers’ strategy significantly reduced disease severity (*p* < 0.001) and incidence (*p* < 0.001), with average values equal to 0.5% and 1%, respectively. The farmers’ plots also significantly differed from the EPI ones in terms of I%I (*p* = 0.005) and I%D (*p* = 0.003) (Figure 5).

### 2.2. Season 2021

#### 2.2.1. Weather Conditions

In 2021, weather conditions in the nine vineyards showed fewer marked differences than in 2020 (Figure 6). The minimum and maximum temperatures, although with values that slightly differed, showed the same trend. In general, the 2021 season was drier than that of 2020; most of the rain was registered in May (58 mm), whereas the period between the 10th of June and the end of July was characterised by an almost total absence of rainfall (Figure 6C), along with a strong increase in temperatures (Figure 6A,B).

#### 2.2.2. Downy Mildew

Overall, the model predicted a high infection risk at three moments: from the 10th to the 24 May, on the 7 June, and on the 12 July (Figure 7).

In most of the vineyards, the treatments in the EPI plots were applied from the 18th of May to the 22nd of June, whereas in the growers’ plots, the sprayings were applied between May 24th and July 6th (Table 3. Even if no statistical differences were found between the two strategies in terms of the number of treatments (*p* = 0.489), more sprayings were applied on average in the FARM plots than in EPI plots; the fungicides were applied seven times in the FARM treatments and six times in the EPI plots (Table 3).

The first downy mildew symptoms appeared between the 7th and 21st of June (Figure 7), caused by infections which probably occurred in the period from the end of May and the first 10 days of June, in accordance with the prediction of the EPI model. In the untreated plot, the average I%D values on leaves and bunches were 22% and 10%, respectively. The I%I in the unsprayed plot was even lower, with average values equal to 4.5% for both leaves and bunches (Figure 8A,B). Disease incidence and severity in the EPI plots were significantly lower than in the untreated ones on leaves (*p* = 0.001) (Figure 8A,C) and bunches (*p* < 0.001) (Figure 8B,D). Following the growers’ schedules, the disease level significantly differed from that registered in the NT parcels for both leaves (0.008 < *p* < 0.005) (Figure 8A,C) and bunches (0.042 < *p* < 0.024) (Figure 8B,D); the average I%I values reached 1.7% on leaves and 2% on bunches, whereas the I%D values reached 10.6% on leaves and 4.6% on grapes. These values were higher than those in the plots sprayed according to the indications of the EPI model, where the average I%I values reached 0.9% on leaves and 0.7% on bunches, whereas the I%D values reached 7.2% on leaves and 2.7% on bunches. However, no significant differences were observed by comparing the two treated plots for both I%D and I%I on leaves (0.560 < *p* < 0.473) (Figure 8A,C) and bunches (0.245 < *p* < 0.105) (Figure 8B,D).

#### 2.2.3. Powdery Mildew

In 2021, the epidemiological model predicted a low-to-medium infection risk for the entire season, except for 2 weeks: the 14th of June (for P2, P7, and P9) and the 12th of July (Figure 9).

During the season, on average, six fungicide sprayings were applied in the EPI strategy and eight in the growers’ strategy (Table 4). The statistical analysis highlighted significant differences between the two strategies in terms of the number of sprayings (*p* = 0.019), where the FARM plots were treated with two more sprayings than were the EPI plots on average.

The first disease symptoms appeared between the 25th of May and the 28th of June, when the model signalled a medium risk in most of the vineyards (Figure 9). A late appearance of the symptoms was observed in some vineyards (P4 and P8). Overall (Figure 10), the disease incidence and severity on leaves (0.234 < *p* < 0.183) and bunches (0.224 < *p* < 0.201) did not significantly differ in the EPI and the NT plots. However, a heterogeneous disease pressure was observed in the different fields (Figure 11): three vineyards (P1, P2, and P5) showed very high percentage values of disease severity (>40% on leaves and >60% on bunches) in the untreated plots. No, or very low (I%I < 1%), disease symptoms were observed in P3, P4, and P7, while in the remaining vineyards (P6, P8, and P9), the I%I values ranged from 1 to 10% on both or at least on one of the two organs (Figure 11). Significant differences were found between I%I (*p* < 0.027 on leaves, *p* < 0.047 on bunches) and I%D (*p* < 0.001 on leaves, *p* < 0.026 on bunches) of the untreated and treated plots in P1, P2, P5, P6, P8, and P9, but not in the remaining vineyards (*p* > 0.07) (Figure 11).

### 2.3. Amounts and Costs of Fungicides

The average quantity of copper (kg/ha) and the total costs of downy mildew treatments performed in the FARM plots were not statistically different from those of the EPI plots (*p* = 0.666) (Table 5). However, for powdery mildew, the treatments costs and the amount of sulphur were significantly higher for the FARM than the EPI plots (*p* = 0.003) (Table 5). Compared to the average amount of sulphur applied in the growers’ spraying program, the quantity of kg/ha was reduced by 21.22%. The farmers’ spraying program was the most expensive control strategy; in fact, the spraying programs based on EPI indications allowed for a savings of 18.63% of the disease management costs.

## 3. Discussion

In the present study the effectiveness of the EPI model in predicting grapevine downy and powdery mildew infection risks was evaluated over a 2-year period in nine vineyards located in Panzano in Chianti (FI), an organic farming district located in Tuscany (Central Italy).

The model’s performance was first validated by comparing simulated and real disease epidemics. Concerning downy mildew management, good efficacy was achieved by following the EPI model indications in the nine vineyards over both growing seasons. The most encouraging results emerged from the prediction of the primary infections and the need for the first fungicide treatment, which represents a key challenge in disease management and is essential in order to avoid the establishment of the disease in the vineyard. The estimation of the incubation period revealed a strong accuracy of the model in predicting the primary infection occurrence. The same occurred for powdery mildew, where the symptoms’ appearance generally followed the risk of infection signalled by the model, apart from a few cases where the disease was observed one month later (P2 in 2020, P4 and P8 in 2021). Since the duration of the infection period cannot be precisely estimated for powdery mildew [14,42], we cannot conclude whether the model overestimated the infection risk at the beginning of the seasons in these vineyards, or whether the incubation period was longer than that of the other vineyards. Early prevention is essential for good control over grape mildew diseases, particularly in organic farming, where no curative products are available [43]. The relatively low average of I%I and I%D values observed in the untreated plots highlighted that the natural disease pressure was very low in both growing seasons. Indeed, the meteorological data analysis revealed that the two years were very similar, with relatively low temperatures at the beginning of the grapevine growing season (May), followed by very high temperatures at the phenological stages of flowering and bunch development (June), with a concomitant absence of rainfall, which resulted in unfavourable conditions for disease development. Therefore, further evaluation of the reliability of the EPI forecasting model under severe disease epidemic conditions is necessary. Indeed, increasing data availability could lead to advances in research on grapevine disease forecasting. As these efforts move towards new tools (such as artificial intelligence and machine learning techniques), it is very important to understand well how the model works well, in particular, in which situations. The prediction of grapevine diseases is a valuable objective because the incidence and the epidemic trend can vary substantially from year to year and from place to place. Moreover, the possibility of predicting these two diseases opens the way to applying rational interventions with fungicide sprays, with a reduction in the environmental impacts. The accuracy of these predictions could have a direct effect on the cost-effectiveness of disease management. Indeed, disease forecasting models are potentially an important part of the future for economically viable crop protection decisions. Furthermore, it is clear that the models need to be locality-specific with adaptations required before their application to an area. Future investigations could involve the use of the model in several different viticultural areas, with the aim of evaluating its real versatility and grower acceptance, which represent the factors which determine the success of a disease forecasting model.

The validation of a disease forecasting model also needs to consider the efficiency in disease control that derives from its utilisation. For this purpose, field assays were carried out in three plots to quantify the disease severity and incidence for both downy and powdery mildew, to evaluate the overall performance of the strategy, and to ensure that the production was not compromised. Analogous levels of downy mildew control were achieved by the two treatment strategies (EPI and FARM), suggesting that the indications provided by the epidemiological model allowed it to achieve an adequate level of disease protection. In both growing seasons, the grower strategy significantly reduced the powdery mildew intensity, compared with the untreated plots. Considering the average disease intensity of the nine vineyards, no significant differences were found between the NT and the EPI plots. However, it must be pointed out that the meteorological conditions of both 2020 and 2021 were poorly conducive to the diseases, as testified by the low average values of disease incidence (14% in 2020 and 33% in 2021) and severity (9% in 2020 and 18% in 2021) recorded in the untreated plots. The very low natural disease intensity did not allow us to appreciate the existence of differences with the EPI strategy that would probably have been seen in the presence of higher disease pressure. Indeed, looking at the P1, P2, and P5 vineyards in 2021, it is possible to note that, in the presence of high disease pressure in the untreated plot, the differences in the disease incidence and severity are clearly visible. Furthermore, since the experimental activities were carried out in commercial vineyards, the untreated and EPI plots were placed close to each other to facilitate the farmers’ management of the field activities. This probably led to disease spread from the untreated to the EPI plots that was particularly marked in P5 during 2021, when the disease intensity in the absence of disease management was greater than 80% and reached 50% in the EPI plot.

Finally, the treatment schedule applied according to the model indications was compared with a standard spraying schedule [44]. Compared to the average number of treatments conducted with the farmers’ strategy, the adoption of the EPI forecasting model resulted in a significant reduction of the number of treatments performed against downy mildew (14% in 2020 and 12.5% in 2021) and, more markedly, against powdery mildew (57% in 2020 and 25% in 2021). The potential reduction of treatments applied for grapevine powdery and downy mildew control is particularly important when compared to the current practises observed in Italy, where growers typically control the diseases by fixed-interval fungicide applications, or by using a calendar-based fungicide spraying program, which leads them to perform more than 10 fungicide sprayings per season against each disease [34,45]. For example, in an organic vineyard located in Veneto (North-Eastern Italy) in 2020, 12 treatments were performed to control downy mildew and 13 to manage powdery mildew [46]. Periodic treatment applications performed at fixed intervals frequently lead to unneeded sprayings, but a common error in powdery mildew control is to delay fungicide application until the disease has become evident in the vineyard. After the disease initiation, the epidemic is generally well-established and more difficult to control [47]. Therefore, the date of the first treatment represents a major lever to significantly decrease fungicide use in vineyards [16] for both diseases. The encouraging results obtained in the current work show that in Tuscany (West-Central Italy), the adoption of the epidemiological model allowed for a decrease of more than 50% in the number of treatments that are commonly performed against powdery and downy mildew [46]. In detail, the adoption of the EPI model resulted in 77% and 54% reductions in the number of sprayings for powdery mildew, and in a reduction of 50% and 42% for downy mildew, in 2020 and 2021, respectively. Our results are consistent with previous studies conducted in other major vine-producing countries. Several disease prediction models were indeed developed for identifying periods when conditions are favourable for grapevine downy and powdery mildew development, and for scheduling necessary fungicide applications [16]. It was demonstrated that the diseases could be controlled effectively with fewer fungicide applications than included in the growers’ schedule by using the forecasting model. For example, the warning system developed by Caffi and co-workers [34,45] reduced fungicide applications by 36% (with a low-risk program) and 75% (with a high-risk program) for powdery mildew control and from 33 to 86% for downy mildew control, compared to standard schedules in Italian vineyards. Similar results were obtained by Pellegrini et al. [25] and Menesatti et al. [48]. Based on results obtained in the current study, it appears that the adoption of the disease forecasting model can reduce spray frequency (for both downy and powdery mildews), and the amount of fungicide applied (for powdery mildew), achieving a good level of disease control. Each saved application represents a reduction of 4.9 L/ha in fuel consumption and a related reduction of about 12.5 kg/ha of CO_2_ emissions [49]. In addition to the environmental benefits, the timely application of fungicide sprayings reduced the costs of production since the saving of fungicide products amounted to EUR 20/ha for downy mildew and to EUR 104/ha for powdery mildew. The comparison of the costs achieved within this study with other studies is quite difficult due to the great variation in human labour costs occurring between countries. Considering powdery mildew, the costs achieved with the model are in line with those of other decision-support systems [50]. Overall, the use of the model allowed for 5% and 19% cost savings for downy and powdery mildew control, and that will definitely help growers facing issues related to the cost of disease management.

## 4. Materials and Methods

### 4.1. Vineyards and Meteorological Data Collection

The experimental assays were carried out over 2 consecutive years (2020–2021) in nine organic vineyards of Panzano in Chianti, in the province of Florence, located in Tuscany (Central Italy) (Figure 12). The Sangiovese variety with the Guyot trellising system (spacing 2.20–2.50 m × 0.7–0.8 m) was grown in all of the vineyards. The Chianti area, where one of the most prestigious red wines is produced (Chianti Classico DOCG), covers an area of approximately 600 km^2^ and is characterised by a predominantly hilly topography. This area was chosen since it shows a high variability in terms of topographical aspects, microclimates, and landscape complexity [51], and therefore, it is a suitable case-study location for model validation. The nine farms (P1–P9) are representative of the context in terms of size, origin, organisation, and management. The vineyards were chosen in order to create a representative spatial distribution within the area, and also in terms of potential susceptibility to downy and powdery mildews.

To increase the spatial and temporal resolution of the weather data, which are of fundamental importance for the adoption of weather-driven disease forecasting models [52], weather sensors and interpolating weather grids were implemented in a spatially wide network using wireless repeaters (Netsens, Calenzano, Italy). Sensors were installed in the vine rows to measure the real microclimatic parameters, and the recorded data were transmitted by IoT wireless units: real-time data gathered from the sensors are transmitted using TCP/IP technology and accessed worldwide through the Netsens LiveData Cloud platform, accessible by any mobile or desktop device. Weather stations were equipped with rain collectors as well as air temperature, humidity, and leaf-wetness sensors (and more). The live data–user interface displays the agrometeorological data from the field in a graphic end-table format through LiveData web-based software (Netsens).

Real and forecasted hourly values for rainfall (mm), minimum and maximum temperatures (°C), and relative humidity (%) were automatically provided to the Epicure system for infection risk simulation with the EPI model.

### 4.2. EPI Model Simulation

Simulations with the EPI model (https://oadex-viti.vignevin-epicure.com/ accessed on 13 April 2020) were carried out twice per week for each vineyard. The levels of disease infection risk were determined according to two indices provided by the model: the FTA (theoretical frequency of attack), which indicates the occurrence of infections) and the EPI index (Etat Potential d’Infection), which indicates whether a pathogen is at a state of high, medium, or low potential risk of infecting. The risk of infection was signalled with a traffic light, where a red light indicated a high risk of infection (strong FTA increase and high EPI index level), a yellow light indicated a medium risk (weak FTA increase and medium EPI index level) and a green light indicated an absence of risk (no FTA increase and low EPI index level).

### 4.3. Field Trials

Each spring, three plots were prepared in each vineyard: the first plot (NT) was not treated against downy or powdery mildews; the second one (EPI) was treated with fungicides targeting downy and powdery mildews according to the indications of the EPI model; the third plot (FARM) was managed by the grower without considering the model indications. Each plot consisted of three replicates of 20 plants. For technical reasons, the NT and EPI plots were placed within the same rows. In the EPI plot, sprays were applied when the model indicated medium (yellow traffic light) to high (red traffic light) risks. Treatments with commercial products containing copper (heliocuivre, 400 g L^−1^ a.s.; Biogard, Grassobbio (BG), Italy), and sulphur (heliosoufre, 700 g L^−1^ a.s.; Biogard Grassobbio (BG), Italy) were carried out with a sprayer pump according to label doses. The treatment schedules performed in the EPI and FARM plots were recorded.

### 4.4. Disease Assessment

Starting with the grapevines’ receptivity to the pathogens, the vineyards were inspected weekly to detect the first appearances of downy and powdery mildew symptoms in the untreated plots. The disease assessment was made at the phenological stage of bunch closure (the end of downy mildew receptivity). For the first year of the study, disease incidence and severity were assessed on bunches, with the aim of evaluating the model’s efficiency in maintaining grapevine production. During the second year, the disease intensity was evaluated on both bunches and leaves in order to consider both yield and quality losses. Inside each plot, three subplots consisting of 15 plants were determined to assess disease on 50 bunches and 100 leaves, which were randomly selected. The organs were carefully inspected for symptoms of both of the diseases, and disease incidence (I%D) was calculated as a percentage of affected leaves or bunches. Disease severity was estimated via the calculation of the percentage index of infection (I%I) [53,54]. The average I%D and I%I of the leaves and bunches were calculated from the individual values calculated for the three subplots. Moreover, the length of the incubation period was calculated [55] so as to ascertain the most probable date of downy mildew infection occurrence.

### 4.5. Data Analysis

The SPSS statistical package for Windows, v. 27 (IBM Italia, Milano), was used for all statistical analyses. The existence of significant differences in the disease incidence (I%D) and severity (I%I) among the three plots (FARM, EPI, and NT) were analysed using the non-parametric Kruskal–Wallis test. The non-parametric Mann–Whitney U test was used to compare the two treated plots (FARM and EPI) in terms of the number of treatments performed in the 2020 and 2021 growing seasons. Moreover, in the 2021 season, the economic impact of the treatments was evaluated by comparing the spraying programs on the basis of the costs related to the fungicide application and on the quantity of copper and sulphur applied (kg/ha). The evaluation of the costs, in euros per hectare, included product costs, fuel consumption, amortisation of equipment, and human labour. Costs of the fungicides were EUR 11/l for heliocuivre, and EUR 5/l for heliosoufre. Costs for human labour, fuel consumption, and amortisation of equipment were assumed to be EUR 50/ha per application, as estimated on the basis of average costs incurred in the nine vineyards.

## 5. Conclusions

In conclusion, the results obtained over two growing seasons (2020–2021) indicated that both the model-timed and winegrowers’ schedule of fungicide treatments significantly reduced the downy mildew level compared to the unsprayed controls. Concerning powdery mildew, the very low disease intensity and the spread of the inoculum from the untreated plots did not allow us to fully appreciate the level of disease control achieved in the EPI plot. Furthermore, the economic damage threshold should be evaluated to understand whether the protection costs are sustainable. More importantly, this study demonstrated that it was possible to achieve similar levels of disease control with timely fungicide applications using the disease prediction model compared to the grower treatments schedule without compromising crop health and, consequently, yield or quality. In particular, for the organic system, where only protective products are available, the timeliness of fungicide application is essential. Further evaluations of the EPI forecasting model under severe powdery and downy mildew epidemic conditions are necessary.

## Figures and Tables

**Figure 1 plants-12-00285-f001:**
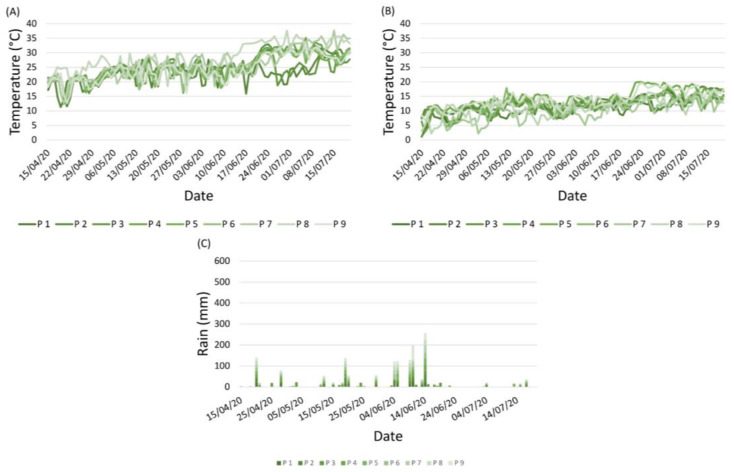
Weather conditions occurring from April to July 2020 in the nine vineyards (P1–P9): daily maximum temperature (**A**), minimum temperature (**B**), and rainfall (**C**). Different shades/colours represent different vineyards (P1–P9).

**Figure 2 plants-12-00285-f002:**
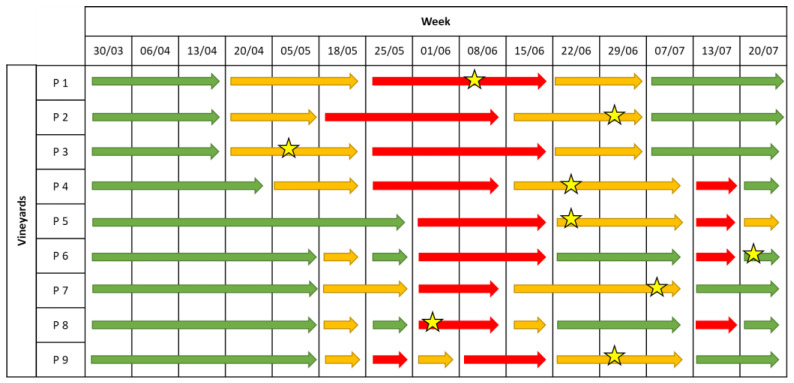
Levels of infection risk signalled by the EPI model. Coloured arrows indicate different levels of infection risk: red indicates a high infection risk, yellow indicates a medium infection risk, and green indicates a low infection risk. The stars indicate the moments when the first symptoms appeared in the field.

**Figure 3 plants-12-00285-f003:**
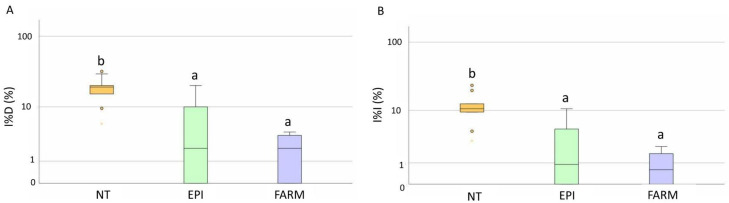
Box plot distribution of the final downy disease incidence (I%D) (**A**) and the severity (I%I) (**B**) registered in 2020 on bunches in the untreated (NT), EPI (EPI), and farmer’s strategy (FARM) plots. Different letters indicate significant differences among samples (*p* < 0.05).

**Figure 4 plants-12-00285-f004:**
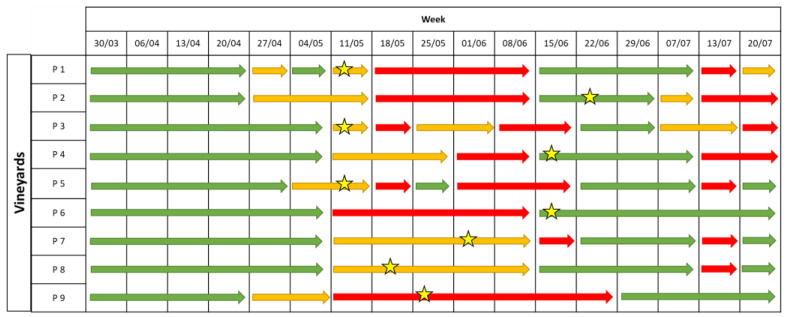
Levels of infection risk signalled by the EPI model. Coloured arrows indicate different levels of infection risk: red indicates a high infection risk, yellow indicates a medium infection risk, and green indicates a low infection risk. The stars indicate the moments when the first symptoms appeared in the field.

**Figure 5 plants-12-00285-f005:**
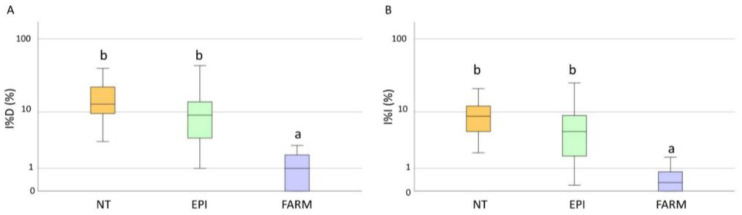
Box plot distribution of the final powdery mildew disease incidence (I%D) (**A**) and severity (I%I) (**B**) registered in 2020 on bunches in the untreated (NT), EPI (EPI), and farmers’ strategy (FARM) plots. Different letters indicate significant differences among samples (*p* < 0.05).

**Figure 6 plants-12-00285-f006:**
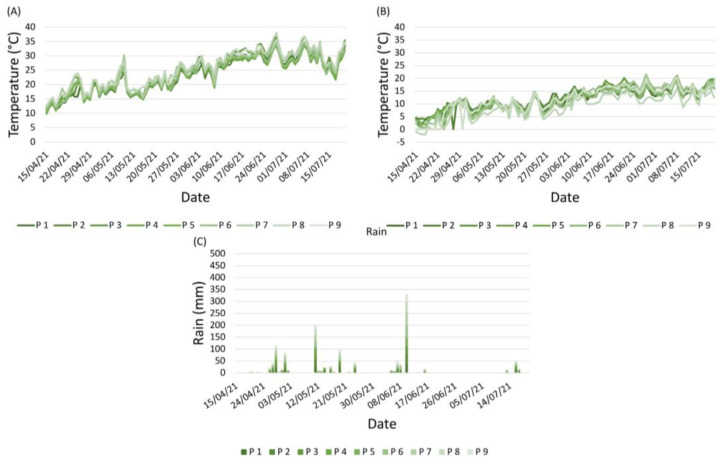
Weather conditions occurring from April to July 2021: daily maximum temperature (**A**), minimum temperature (**B**), and rainfall (**C**) in the nine vineyards (P1–P9). Different shades/colours represent different vineyards.

**Figure 7 plants-12-00285-f007:**
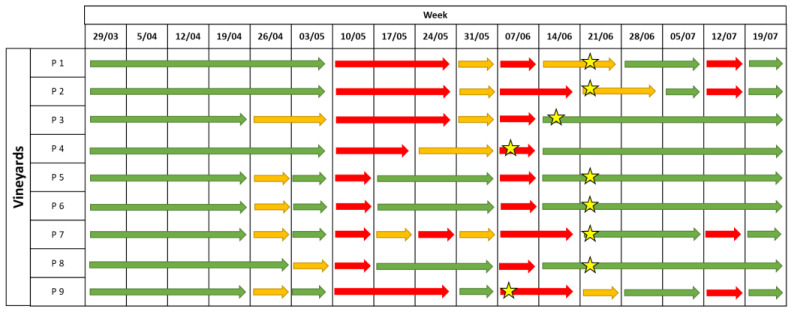
Levels of infection risk signalled by the EPI model. Coloured arrows indicate different levels of infection risk: red indicates a high infection risk, yellow indicates a medium infection risk, and green indicates a low infection risk. The stars indicate the moments when the first symptoms appeared in the field.

**Figure 8 plants-12-00285-f008:**
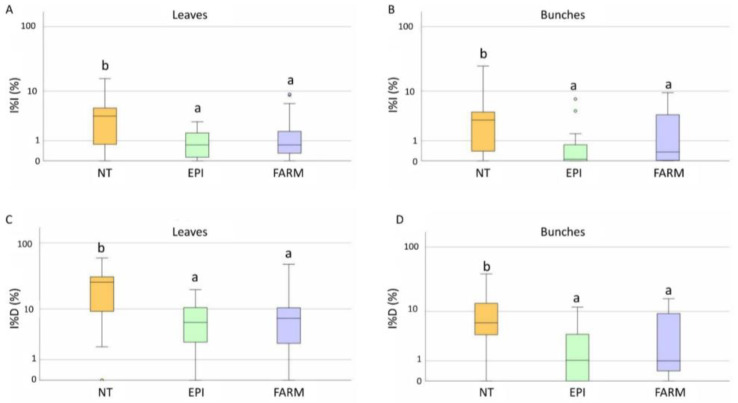
Box plot distribution of the final downy mildew disease severity (I%I) on leaves (**A**) and bunches (**B**), and incidence (I%D) on leaves (**C**) and bunches (**D**) registered during 2021 in the untreated (NT), EPI (EPI), and farmers’ strategy (FARM) plots. Different letters indicate significant differences among samples (*p* < 0.05).

**Figure 9 plants-12-00285-f009:**
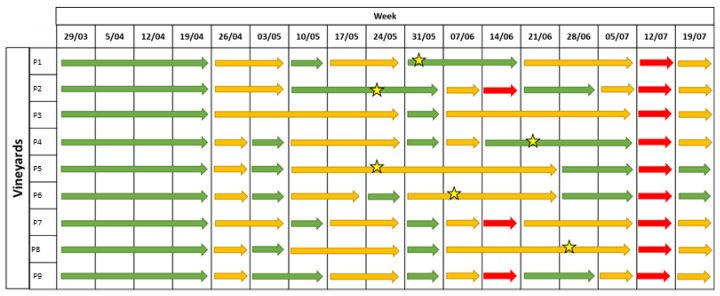
Levels of infection risk signalled by the EPI model. Coloured arrows indicate a different level of infection risk: red indicates a high infection risk, yellow indicates a medium infection risk; and green indicate a low infection risk. The stars indicate the moments when the first symptoms appeared in a field.

**Figure 10 plants-12-00285-f010:**
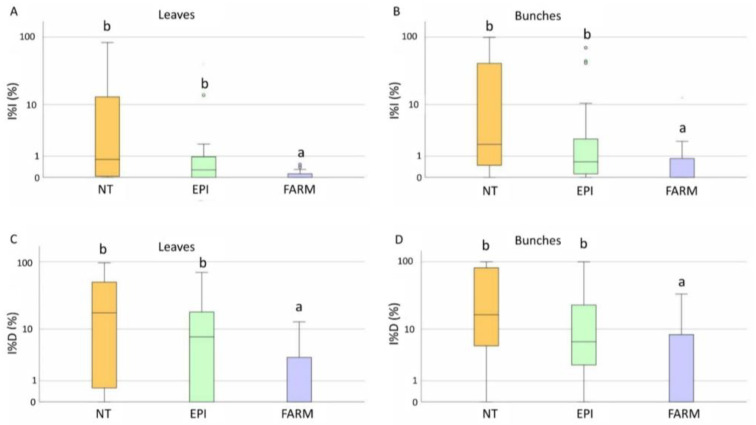
Box plot distribution of the final powdery mildew disease severity (I%I) on leaves (**A**) and bunches (**B**), and incidence (I%D) on leaves (**C**) and bunches (**D**) registered during 2021 in the untreated (NT), EPI (EPI), and farmers’ strategy (FARM) plots. Different letters indicate significant differences among samples (*p* < 0.05).

**Figure 11 plants-12-00285-f011:**
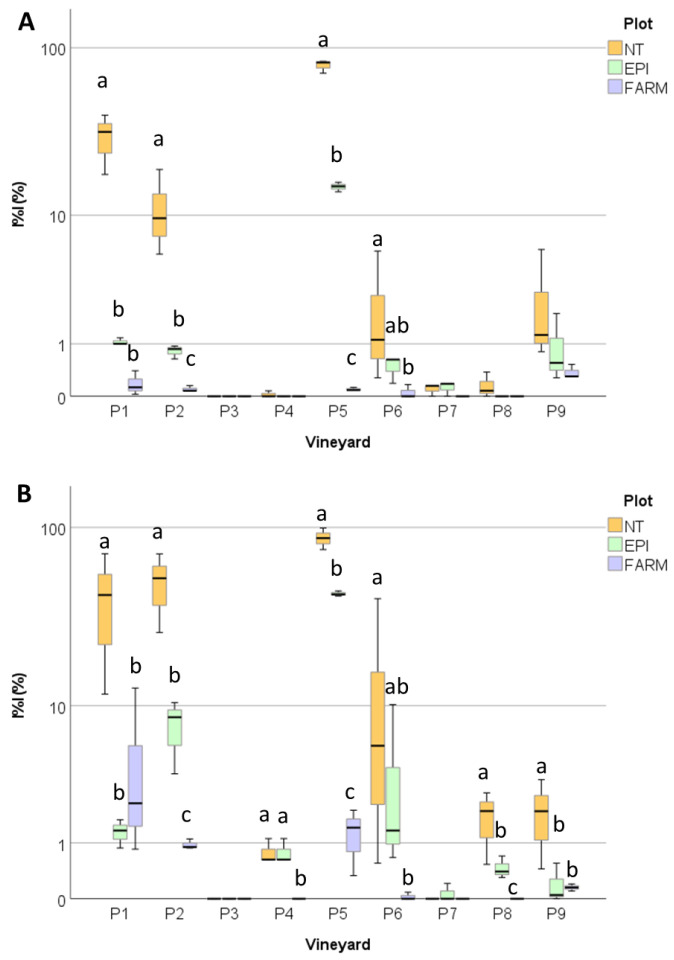
Box plot distribution of the final disease severity on leaves (**A**) and bunches (**B**) registered in 2021 in the untreated (NT), EPI (EPI), and farmers’ strategy (FARM) plots of the nine vineyards (P1–P9). Different letters indicate significant differences among samples (*p* < 0.05).

**Figure 12 plants-12-00285-f012:**
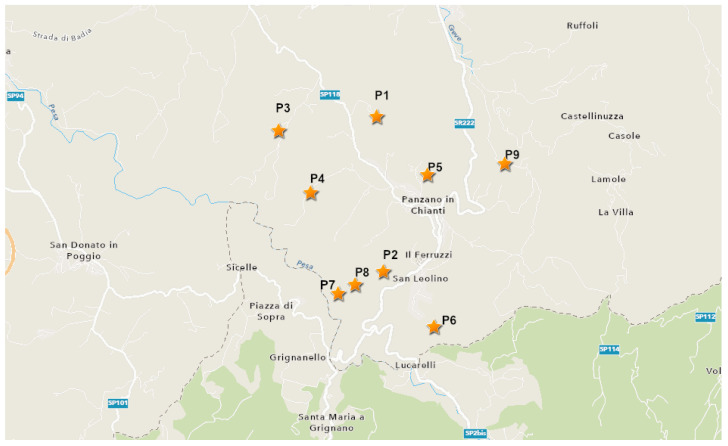
Map of the province of Panzano in Chianti (FI), where the locations of the vineyards (P1–P9) are indicated by orange stars. Map made with ArcGIS (https://www.arcgis.com/home/webmap/viewer.html accessed on 20 November 2022).

**Table 1 plants-12-00285-t001:** Dates of spray applications and total number of treatments performed for downy mildew control in EPI (E) and farmer (F) plots of the nine considered vineyards during 2020. Different letters indicate significant differences among samples (*p* < 0.05).

		Week																					
		14/05		25/05		01/06		08/06		15/06		22/06		29/06		07/07		13/07		20/07		Total	
		E	F	E	F	E	F	E	F	E	F	E	F	E	F	E	F	E	F	E	F	E	F
**Vineyards**	**P1**		X	X	X	X		X	X	X	X	X			X		X		X		X	**5**	**8**
	**P2**		X	X	X	X		X	X	X	X	X			X		X		X			**5**	**7**
	**P3**		X	X	X	X		X	X	X	X		X		X		X		X		X	**4**	**9**
	**P4**				X	X	X	X		X	X	X	X	X	X	X	X		X			**6**	**7**
	**P5**		X	X	X	X	X	X	X	X	X	X	X	X	X			X	X	X	X	**8**	**9**
	**P6**					X	X	X	X	X	X	X	X	X	X	X	X	X		X	X	**8**	**7**
	**P7**				X	X	X	X		X	X	X	X	X	X				X			**5**	**6**
	**P8**			X	X	X	X	X	X	X	X	X	X	X	X	X	X		X			**7**	**8**
	**P9**			X		X	X	X	X	X	X	X	X	X	X	X	X		X	X	X	**8**	**8**
																						**Average**	
																						**6a**	**8b**

**Table 2 plants-12-00285-t002:** Dates of spray applications and total number of treatments performed for powdery mildew control in EPI (E) and farmer (F) plots of the nine studied vineyards during 2020. Different letters indicate significant differences among samples (*p* < 0.05).

		Week																					
		14/05		25/05		01/06		08/06		15/06		22/06		29/06		07/07		13/07		20/07		Total	
		E	F	E	F	E	F	E	F	E	F	E	F	E	F	E	F	E	F	E	F	E	F
**Vineyards**	**P1**		X		X		X	X	X	X	X	X	X		X		X					**3**	**8**
	**P2**		X		X		X	X		X		X	X		X		X					**3**	**6**
	**P3**		X		X		X	X	X	X	X	X	X		X		X		X			**3**	**9**
	**P4**						X		X	X	X	X	X		X		X					**2**	**6**
	**P5**		X		X		X		X	X	X	X	X		X		X		X	X		**3**	**9**
	**P6**				X		X			X	X	X	X	X	X				X		X	**3**	**7**
	**P7**				X		X	X	X	X	X	X	X				X					**3**	**6**
	**P8**				X		X	X	X	X	X	X	X		X		X					**3**	**7**
	**P9**				X		X	X	X	X	X	X	X	X	X		X		X			**4**	**8**
																						**Average**	
																						**3a**	**7b**

**Table 3 plants-12-00285-t003:** Date of spray applications and total number of treatments performed for downy mildew control in EPI (E) and farmer (F) plots of the nine vineyards during 2021. Different letters indicate significant differences among samples (*p* < 0.05).

		**Week**																							
		**27/04**		**04/05**		**10/05**		**18/05**		**24/05**		**01/06**		**08/06**		**15/06**		**22/06**		**29/06**		**06/07**		**Total**	
		**E**	**F**	**E**	**F**	**E**	**F**	**E**	**F**	**E**	**F**	**E**	**F**	**E**	**F**	**E**	**F**	**E**	**F**	**E**	**F**	**E**	**F**	**E**	**F**
**Vineyards**	**P1**			X		X	X	X		X	X	X	X	X	X	X	X	X	X					**8**	**6**
	**P2**					X	X	X		X	X	X		X	X	X		X	X	X	X			**8**	**5**
	**P3**							X		X	X	X	X	X	X	X	X	X	X		X		X	**6**	**7**
	**P4**					X	X	X		X	X	X	X	X	X				X		X		X	**5**	**7**
	**P5**					X	X			X	X			X	X			X	X	X	X			**5**	**5**
	**P6**							X		X	X	X	X	X	X	X	X	X	X	X	X		X	**7**	**7**
	**P7**							X		X	X	X	X	X	X	X	X	X	X		X		X	**6**	**7**
	**P8**								X	X	X			X	X	X	X	X	X		X		X	**4**	**7**
	**P9**							X	X	X	X	X	X	X	X	X	X	X	X		X		X	**6**	**8**
																								**Average**	
																								**6a**	**7a**

**Table 4 plants-12-00285-t004:** Dates of spray applications and total number of treatments performed for powdery mildew control in EPI (E) and farmers’ (F) plots of the nine vineyards during 2021. Different letters indicate significant differences among samples (*p* < 0.05).

		Week																					
		27/04		10/05		18/05		24/05		01/06		08/06		15/06		22/06		29/06		06/07		Total	
		E	F	E	F	E	F	E	F	E	F	E	F	E	F	E	F	E	F	E	F	E	F
**Vineyards**	**P1**	X		X	X	X			X		X		X		X	X	X	X	X	X	X	**6**	**8**
	**P2**			X	X		X		X		X	X	X		X	X	X	X	X	X	X	**5**	**9**
	**P3**			X		X	X					X	X	X	X	X	X	X	X	X	X	**7**	**6**
	**P4**			X		X	X	X	X	X	X	X	X		X	X	X	X	X	X	X	**8**	**8**
	**P5**			X	X	X	X		X		X	X	X	X	X	X	X	X	X	X	X	**7**	**9**
	**P6**			X					X		X	X	X		X	X	X	X	X	X	X	**5**	**7**
	**P7**			X		X	X				X	X	X			X	X	X	X	X	X	**6**	**6**
	**P8**			X	X		X				X	X	X	X	X	X	X	X	X	X	X	**6**	**8**
	**P9**			X		X	X		X		X	X	X		X	X	X	X	X	X	X	**6**	**8**
																						**Average**	
																						**6a**	**8b**

**Table 5 plants-12-00285-t005:** Averages and standard deviations of the amounts of copper and sulphur applied in EPI and farmers’ plots and treatments costs for 2021. Different letters indicate significant differences between plots (*p* < 0.05).

Disease	Parameter	Plot	
		EPI	Farm
DM	Copper (Kg/ha)	2.3 ± 0.5 a	2.4 ± 0.5 a
	Costs (EUR/ha)	368.3 ± 81.4 a	388.2 ± 74.9 a
PM	Sulphur (Kg/ha)	19.3 ± 3.3 a	24.5 ± 3.5 b
	Costs (EUR/ha)	454.3 ± 66 a	558.3 ± 81.1 b

## Data Availability

The data presented in this study are available on request from the corresponding author.

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
