# Peer review of "Disease Forecasting for the Rational Management of Grapevine Mildews in the Chianti Bio-District (Tuscany)"

_plants, 2023, doi:10.3390/plants12020285_

Round 1

Reviewer 1 Report

The Paper " Disease forecasting for a rational management of grapevine 2 mildews in the Chianti bio-district (Tuscany)" is describing the efficiency of EPI model in predicting risk of infection downy and 12 powdery mildew. 

This reviewer believes the paper is well written, organized and discussed, and most importantly the objectives and findings of the paper is important for the researchers in the area of viticulture, thus has the merit to be published as its current form.

Author Response

Dear  Reviewer,

on behalf of the authors I wish to thank you for your comments.

All the best,

Silvia Toffolatto

Reviewer 2 Report

The manuscript “Disease forecasting for a rational management of grapevine mildews in the Chianti bio-district (Tuscany)” is important, because brings new information that lead to advances in research on grapevine disease forecasting. This approach can permit reduce the number of spraying at the crops and the cost with fungicides to control those important diseases in grapevine. I made few suggestions and corrections at the text. I think it is good, but I have a suggestion for the authors: As the model permited to reduce the number of the spraying, for me could be good to present an economic discussion about it. How many money the farmers can save using this model? 

Author Response

Dear Reviewer,

many thanks for your precious comments that contributed to the manuscript improvement.

We inserted "Etat Potentiel d'Infection" in the abstract, as suggested in the pdf file.

Concerning the economic analysis, we pointed out the costs savings achieved in this study at the end of the discussion as follows: "The comparison of the costs achieved within this study with other studies is quite difficult, due to the great variation in the human labor costs occurring among countries. Considering powdery mildew, the costs achieved with the model are in line with those of other decision support systems [50]. Overall, the use of the model allowed 5 % and 19% cost savings for downy and powdery mildew control that will definitely help the growers in facing the issues related with the costs of the diseases management."

Many thanks again and best regards by the authors of the work.